# Regional and demographic variations of Carotid artery Intima and Media Thickness (CIMT): A Systematic review and meta-analysis

**V. Abeysuriya** [1,2]*, **B. P. R. Perera**[1], **A. R. Wickremasinghe**[1]

**1** Department of Public Health, Faculty of Medicine, University of Kelaniya, Ragama, Sri Lanka, **2** Nawaloka Hospital Research and Education Foundation, Nawaloka Hospitals PLC, Colombo, Sri Lanka

* visulasrilanka@hotmail.com

## Abstract

### Background and objective

Carotid artery intima media thickness (CIMT) is a strong predictor of Coronary Heart Disease (CHD) and independent phenotype of early atherosclerosis. The global variation of CIMT and its demographic association is yet unclear. We evaluated regional variations of CIMT based on WHO regions and assessed the differences by age and sex.

### Methods

A systematic search was conducted on studies published between 1980 January up to December 2020. PubMed, Oxford Medicine Online, EBSCO, Taylor & Francis, Oxford University Press and Embase data bases were used for searching. Supplementary searches were conducted on the Web of Science and Google Scholar. Grey literature was searched in "Open Grey" website. The two major criteria used were "adults" and "carotid intima media". The search strategy for PubMed was created first and then adapted for the Oxford Medicine Online, EBSCO, Taylor & Francis, Oxford University Press and Embase databases. Covidence software (Veritas Health Innovation, Melbourne, Australia; http://www.covidence.org) was used to manage the study selection process. Meta-analyses were done using the random-effects model. An $I^2 \geq 50\%$ or $p < 0.05$ were considered to indicate significant heterogeneity.

### Results

Of 2847 potential articles, 46 eligible articles were included in the review contributing data for 49 381 individuals (mean age: 55.6 years, male: 55.8%). The pooled mean CIMT for the non-CHD group was 0.65mm (95%CI: 0.62–0.69). There was a significant difference in the mean CIMT between regions ($p = 0.04$). Countries in the African (0.72mm), American (0.71mm) and European (0.71mm) regions had a higher pooled mean CIMT compared to those in the South East Asian (0.62mm), West Pacific (0.60mm) and Eastern Mediterranean (0.60mm) regions. Males had a higher pooled mean CIMT of 0.06mm than females in the non CHD group ($p = 0.001$); there were also regional differences. The CHD group had a significantly higher mean CIMT than the non-CHD group (difference = 0.23mm, $p = 0.001$) with

**Data Availability Statement:** All data files are available from the figshare database (Abeysuriya, Visula (2022): CIMT Dataset. figshare. Dataset. https://doi.org/10.6084/m9.figshare.19360925.v1).

**Funding:** VA received the funds from Nawaloka Hospital Research and Education Foundation, Nawaloka hospitals PLC, Colombo-02 Sri Lanka. Grant number is NHREF/01/2020. Funder web site: https://www.nawaloka.com. The funders had no role in study design, data collection and analysis, decision to publish, or preparation of the manuscript. the funder provided support in the form of salaries for the author [VA] but did not have any additional role in the study design, data collection and analysis, decision to publish, or preparation of the manuscript. The specific roles of these authors are articulated in the 'author contributions' section.

**Competing interests:** The authors have declared that no competing interests exist.

regional variations. Carotid artery segment-specific-CIMT variations are present in this population. Older persons and those having CHD group had significantly thicker CIMTs.

## Conclusions

CIMT varies according to region, age, sex and whether a person having CHD. There are significant regional differences of mean CIMT between CHD and non-CHD groups. Segment specific CIMT variations exist among regions. There is an association between CHD and CIMT values.

## Introduction& rationale

The global burden of non-communicable diseases (NCD)varies between developed and developing countries showing regional differences [1–4]. NCDs are the leading cause of death and disability worldwide. In 2005, NCDs caused an estimated 35 million deaths comprising 60% of all deaths globally; 80% of these deaths were in low income and middle-income countries [5, 6]. NCDs are inextricably linked to many modifiable and non-modifiable risk factors [1, 7–9]. Coronary heart disease (CHD) is the leading cause of premature deaths [10–12].

An accurate, non-invasive, convenient and low-cost screening tool to detect CHD is needed for mass screening of at-risk population. The Carotid intima-media thickness (CIMT) is a reliable, non-invasive indicator which predicts the risk of coronary artery disease (CAD) and is widely used in practice as an inexpensive, reliable, non-radiation and reproducible method [13–19].

CIMT is mostly associated with traditional cardiovascular risk factors such as age, sex and race [20–22]. Smoking, alcohol consumption, lack of exercise, high blood pressure, dyslipidemia, poor dietary patterns, risk-lowering drug therapy, glycemia, hyperuricemia, obesity-related anthropometric parameters and obesity-related diseases increase CIMT [23–25]. Traditional risk factors do not explain all of the risk of CHD. It has been reported that more than 60% of CHD cases were not explained by demographic and traditional cardiovascular risk factors [26]. This may probably be due to the effects of novel risk factors such as heredity, presence of certain genotypes, immunological diseases, inflammatory cytokines and hematological parameters [27–30].

Majority of research on CIMT and its association with future risk of cardiovascular disease (CVD) independent of conventional risk factors has been done in Western populations. Only one study has been conducted in Asia in a Japanese population with a limited sample size [31]. Literature suggests that using CIMT cut-off values of western populations for risk prediction of Asians may not be appropriate [32]. CIMT values are strongly affected by age, sex and population [33]. Therefore, CIMT cut-offs are needed for its clinical use as a screening tool to predict future cardiovascular risk [33]. The manner in which CIMT is assessed and the definitions used are still not universally defined [16, 34, 35].

It is not possible to review CIMT values for each country as such values are not available for many countries. Therefore, we reviewed available literature by WHO region, assuming that populations within the region are more homogenous, to derive potential CIMT cut-off values by age and sex that may be used by different countries in the regions.

## Method and analysis

We followed guidelines of Preferred Reporting Items for Systematic Reviews and Meta-Analyses (PRISMA) statements, the Meta-analysis of Observational Studies in Epidemiology (MOOSE) guidelines, and methods outlined in the Cochrane Handbook for Systematic Reviews of Interventions [36–38] to conduct this review and meta-analysis.

### Eligibility criteria

**Study designs.**   Studies of observational and interventional research were included. The following study designs having adults with a mean age of 40 years and above, with or without CHD were considered: longitudinal, case–control, nested case-control and cross-sectional studies. Case reports, case series, opinion papers, letters to the editor, comments, conference proceedings, review articles, policy papers and meta-analyses were excluded from the analysis. Animal studies, non-English manuscripts and study protocols without baseline data were excluded. The outcome measure was the intima-media thickness of the carotid artery measured by ultrasonography. There was no restriction by time duration of follow-up or observation.

**Setting.**   Data from all countries were considered. There was no restriction by type of setting. The countries were later categorized into WHO regions.

The six WHO regions are 1) African Region (AFR); 2) Eastern Mediterranean Region (EMR); 3) European Region (EUR); 4) Region of the Americas (PAHO); 5) South-East Asia Region (SEAR); and 6) Western Pacific Region (WPR) (40).

**Search strategy.**   Potential articles were systematically searched in the following electronic databases; PubMed, Oxford Medicine Online, EBSCO, Taylor & Francis, Oxford University Press and Embase for publications between January 1980 to December 2020. Supplementary searches were done on Web of Science and Google Scholar. Grey literature was searched in "OpenGrey" website using two criteria "adults" and "carotid intima media". The search strategy for PubMed was created first and then adapted for the Oxford Medicine Online, EBSCO, Taylor & Francis, Oxford University Press and Embase data bases (S1 File). The references of these selected articles were hand-searched for more relevant articles.

**Study selection.**   After removing duplicates and obviously unrelated articles, the titles and abstracts were screened against pre-specified criteria by two independent reviewers. Pre-determined inclusion criteria were based on the following key words: "carotid intima media thickness", "coronary heart disease", "healthy adults", "adults with coronary heart disease", and "studies in English language". Exclusion criteria included "children", "paediatric", "any person with a history of stroke or TIA", "history of malignancy", "who has undergone carotid end arterectomy", "history of connective tissue disease", "history of an ongoing infection", "studies on cadaver or corpse", "studies on animals", "other languages", "meta–analysis", "reviews", and "letters to editor". Discrepancies were resolved through discussion. If consensus was not reached, arbitration was done with a third reviewer. Full text articles were assessed for eligibility. The systematic reviews software Covidence (Veritas Health Innovation, Melbourne, Australia; http://www.covidence.org) was used to manage the study selection process.

**Data extraction.**   The following data were extracted: name of first author; year of publication; country (according to WHO regions), study design, number of patients, age, proportion of males and females, number of CHD and non-CHD persons, segment measured, measurement protocol, risk factors, mean and maximum values of CIMT. Two authors independent of each other extracted data. Disagreements were resolved by discussion or, if necessary, with the arbitration of a third reviewer. Calibration exercises were conducted before this review stage to enhance consistency between assessors. The study team collated information provided in

 

multiple reports of the same study. For articles on the same population, the more comprehensive one was selected. Apart from inclusion and exclusion criteria, authors selected studies with adjusted CIMT values and study quality assessment statements were considered. When CIMT measurements were available for several time points, the time point closest to the end of the intervention or the follow-up period was selected for data extraction. When essential information was missing from the published reports, the principal investigator contacted the authors of the original studies by email or through "Research gate" to request for missing data. A maximum of two email attempts per study was made.

**Study quality.** The quality of selected studies was assessed using the Quality Assessment of Diagnostic Accuracy Studies (QUADAS-2) criteria [39], the "STROBE statement" quality assessment tool and "The Newcastle-Ottawa Scale" were used to assess quality and heterogeneity of case control, cross sectional and cohort studies, and risk of bias [40]. Quality appraisal was performed independently by two reviewers. The protocol of ultrasound measurement of CIMT and reliability was assessed based on "A Consensus Statement from the American Society of Echocardiography Carotid Intima-Media Thickness Task Force" [41].

## Data analysis

Data analysis was carried out using STATA version 16 (Stata Corp. 2019. Stata Statistical Software: Release 16. College Station, TX: Stata Corp LLC).

## Measures of association

Differences in CIMT by age, sex and selected risk factors in countries between WHO regions.

**Descriptive analyses.** The characteristics of the study population including details of publication, country, WHO region, age, gender, sample size, measurement site, CIMT assessment, ultrasound protocol and process, identified risk factors, factors adjusted for and adjusted predictors of CIMT in each study are presented in the text and as tables.

## Steps of meta-analyses

The mean CIMT were pooled according to WHO regions. Based on the literature we expected to have heterogeneity between the pooled data [16, 41–44]. Therefore, meta-analyses were done using random-effects models with inverse variance-weighted average. Results are presented graphically as forest plots. Meta regression analysis of CIMT values was conducted with and without adjusting for coronary heart disease status, region, mean age and ultrasound technique used.

**Assessment of heterogeneity of studies.** Heterogeneity was tested using the Cochran's Q test and quantified using the $I^2$ [38]. An $I^2 \geq 50\%$ or $p < 0.05$ was considered as indicating significant heterogeneity [45]. Sensitivity analyses were carried out by excluding studies with relatively small sample sizes and low-quality studies based on the scores of QUADAS-2 criteria, "STROBE statement "and "The Newcastle-Ottawa Scale".

**Assessment of strength of evidence.** Quality Assessment of Diagnostic Accuracy Studies (QUADAS-2) criteria, "STROBE statement"and "The Newcastle-Ottawa Scale" were applied to evaluate the quality of the included articles [39]. QUADAS-2 criteria assess the strength of evidence by categorizing studies into low risk, high risk and unclear based on patient selection, index test, and reference standard, flow and timing domains. The "STROBE statement" checklist consists of 22 items that relate to the title, abstract, introduction, methods, results, and discussion sections of articles. Eighteen items are common to all three study designs and four are specific for cohort, case-control, or cross-sectional studies. The primary outcome was the STROBE score, defined as the number of the 22 STROBE items adequately reported divided

by the number of applicable items, expressed as a percentage [46, 47]. Publication quality grades of STROBE score are as follows: excellent (more than 85%), good (85 to 70%), fair (70 to 50%) and poor (less than 50%). The Newcastle-Ottawa Scale considers study selection, comparability and outcome categories when assessing the quality of selected studies. The points are considered as follows: 4 points for selection, 2 points for comparability, and 3 points for outcomes. Study quality was categorized according to total points obtained by each study (very good [9], good [7–8], satisfactory [5–6] and unsatisfactory [0–4] [48, 49]).

## Results

2847 [(records identified through data bases: n = 2647; published Literature (PL): 2502 (94.5%); grey literature (GL): 145(5.5%) and records identified through other sources (n = 200); PL: 192(96%); GL: 8(4%))] relevant articles were obtained; 93 records were duplicates and were removed (Fig 1). The abstract and titles were screened, and 2201 articles were removed due to different populations, disease outcomes and study designs, other methods of CIMT measurement, animal studies and non-English publications. Full texts of the remaining 553 publications were evaluated for eligibility (n = 553, published literature: 549(99.2%); grey literature: 4(0.8%)). From the review of the full texts, an additional 507 articles were removed due to different study designs, study populations, outcomes and settings, insufficient data and paediatric population. Finally, 46 eligible articles were reviewed [PubMed: 11(23.9%), EBSCO: 9(19.6%), Taylor & Francis 9(19.6%), Embase 7(15.2%), Oxford Medicine6 (13.0%), Oxford University Press 4(8.7%)] (Fig 1).

Two independent reviewers conducted the full text review. The agreement between the two reviewers was 90% with a Cohen's kappa of 0.733. All the studies were evaluated using QUADAS-2, "STROBE statement" and "The Newcastle-Ottawa Scale" for cross sectional, case control and cohort studies, respectively. QUADAS-2 risk of bias and applicability of the selected studies is shown in Fig 2. The percentages of low-risk studies based on patient selection, index test, reference standard and flow and timing domains were 93.5%, 84.7%, 65.2% and 65.2%, respectively. In the applicability category, it was 64.4% for patient selection, 82.7% for index test and 52.2% for reference standard.

91.3% (42/46) of the studies fulfilled the criteria of the STROBE statement (S1 Table). The Newcastle–Ottawa scale was used to assess the quality of selected studies. Average total quality score for Newcastle–Ottawa scale of cross sectional, case control and cohort studies were7, 7 and 8, respectively (S2–S4 Tables, respectively).

Table 1 provides an overview of the 46 studies included in the systematic review and meta-analyses. The studies were categorized based on the countries they were conducted in according to WHO regions: African Region (AFR) had6(13%) studies; Eastern Mediterranean Region (EMR) had 4(8%); European Region (EUR) had 12(26%); Region of the Americas (PAHO) had8(17%); South-East Asia Region (SEAR) had 7(15%); and the Western Pacific Region (WPR) had 9(20%) studies. There were 24(52%) cross sectional studies, 20(445%) case control studies, 01(2%) prospective study and 01(2%) retrospective cohort study included in the systematic review. There was heterogeneity when measuring the CIMT value among the studies. The commonest segment measured was the far wall of the common carotid artery (CCA) (both sides) (19/46 = 41%), followed by the far wall of CCA, carotid bulb (CB) and internal carotid artery (ICA) (both sides) (6/46 = 13%) and the far and near walls of CCA, CB and ICA (both sides) (4/46 = 9%). The most common IMT definition used was mean CIMT (30/46 = 65%). Definition of plaque was reported in 58% of studies (27/46). ECG gating at acquisition was reported in 28%(13/46) of studies. All studies had used a linear transducer with the frequency varying from 3MHz to 15MHz. Only five studies used Digital Imaging and

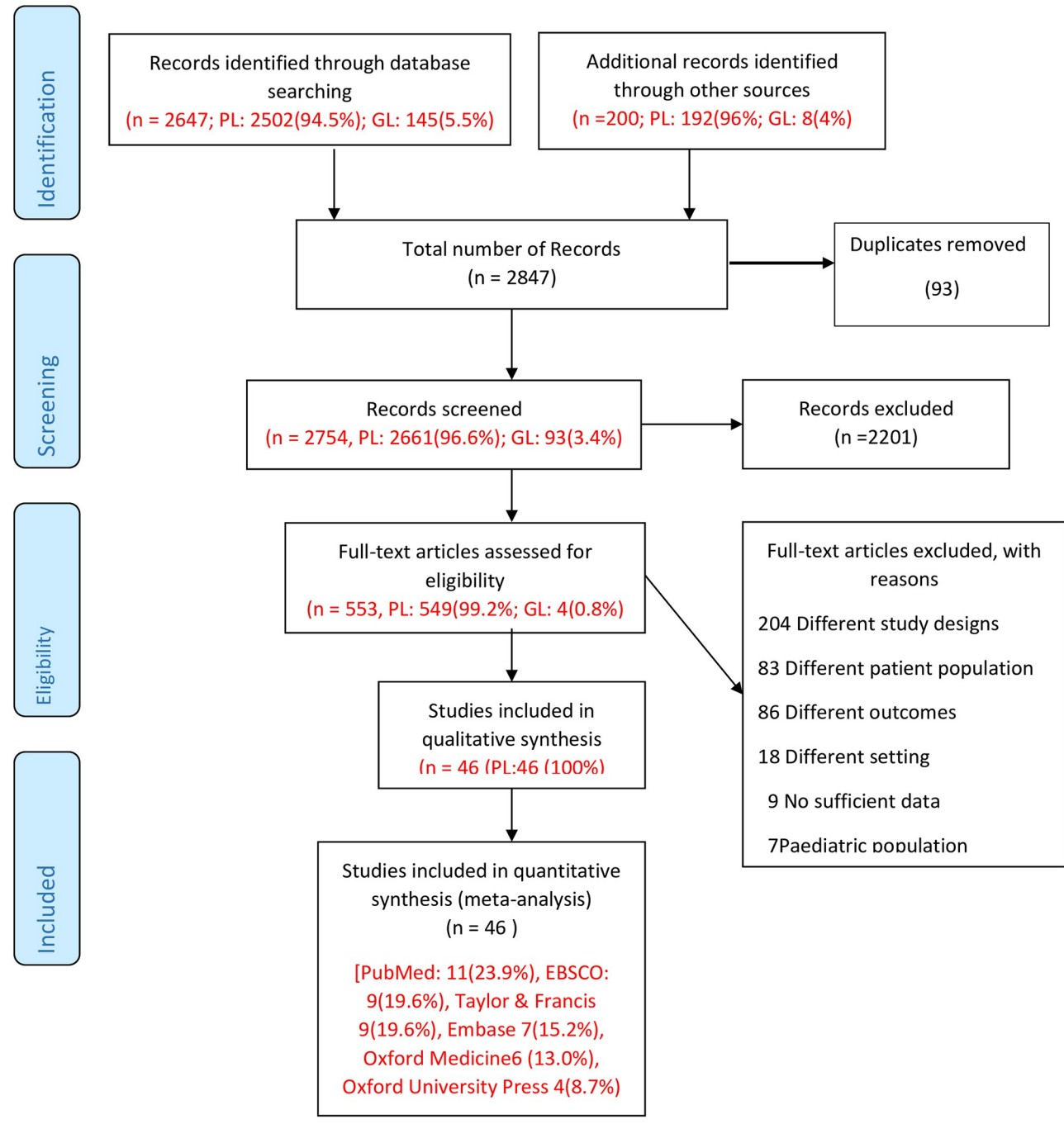

**Fig 1. PRISMA flow diagram.** Abbreviation: PL: publish literature, GL: Grey literature.

Communications in Medicine (DICOM) software. Traditional modifiable risk factors were the commonest predictors of CIMT (21/46 = 45.6%) followed by non-modifiable risk factors of age and gender (13/46 = 28%). Three studies reported age as a single predictor of CIMT (3/46 = 7%). One study reported air pollution as a risk factor for CIMT. Three studies reported socio-economic status as a predictor of CIMT. HIV infection, CRP levels and metabolic

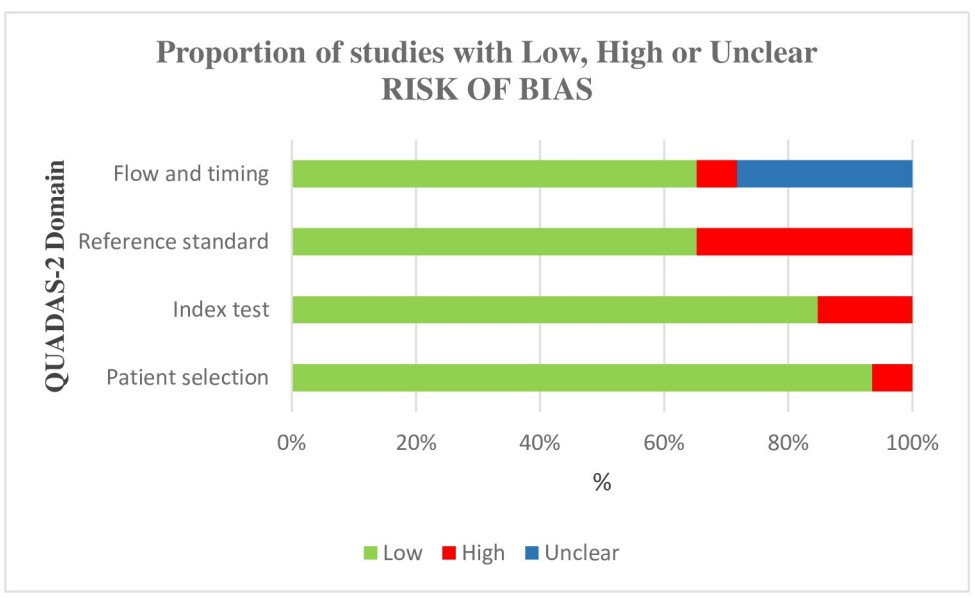

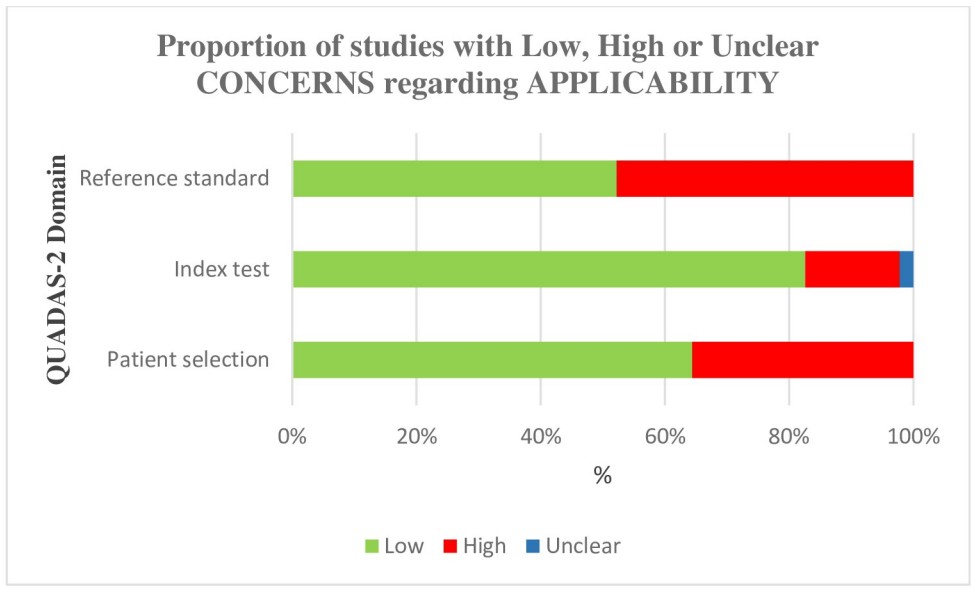

**Fig 2. QUADAS-2 risk of bias and applicability of selected studies.**

syndrome were reported as predictors of CIMT in a few studies. Only one study reported that none of the traditional risk factors predicted CIMT.

Table 2 shows the mean CIMT values of different carotid segments by WHO region in the non-CHD and CHD groups. The mean CIMT values of CCA vary from 0.58±0.09mm to 0.74 ±0.11mm. The mean CIMT of CB ranges from 0.65±0.08mm to 0.81±0.09mm. The range for the mean CIMT of ICA was 0.65±0.10mm to 0.69±0.06. In each region, the highest mean CIMT value was in the CB followed by the CCA and the ICA. The highest mean CIMT value of CCA of 0.74±0.11mm was in EUR countries. The mean CIMT values of CCA in SEAR and

**Table 1. Summary of studies used in systematic review and meta-analysis, reporting demography, IMT measurement protocols and predictors of CIMT.**

| Publication | Country | WHO regions | Design | Sample size | Mean age (years) | Male N, % | Carotid segments | IMT definition | Definition of plaque | Ultrasound scan specifications | ECG gating at acquisition | Factors adjusted for |
|---|---|---|---|---|---|---|---|---|---|---|---|---|
| Denise et al. 2018 [50] | Nigeria | AFRO | Cross-sectional study | 100 | 58.3 | N = 44, 44.0% | CCA, CB, ICA, both sides, far wall | Mean CIMT | Reported | DICOM- Not used, Transducer- Linear:7.5-10MHz, Edge detection: Not used | Not used | Age, gender, smoking, BMI and hypertension |
| Ayoola et al. 2015 [51] | Nigeria | AFRO | Case control study | 100 | 54.9 | N = 50, 50.0% | CCA, both sides, far wall | Mean CIMT | Reported | DICOM- Not used, Transducer- Linear:7.5-10MHz, Edge detection: Not used | Not used | Hypertension, gender, FBS dyslipidemia |
| Ofonime et al. 2019 [52] | Nigeria | AFRO | Cross-sectional study | 122 | 52.7 | N = 36, 29.5% | CCA, CB, ICA, both sides, far wall | Mean CIMT | Not reported | DICOM- Not used, Transducer- Linear:7.5-10MHz, Edge detection: Not used | Not used | Age, DBP, gender, Family history of heart disease, BMI, Physical activity, Waist circumference and SBP |
| Okeahialam et al. 2011 [53] | Nigeria | AFRO | Cross-sectional study | 71 | 50 | N = 35, 49.3% | CCA, both sides, far wall | Mean CIMT | Not reported | DICOM- Not used, Transducer- Linear:7.5MHz, Edge detection: Not used | Not used | Gender, Diabetes and hypertension |
| Zaiboonnisa et al. 2009 [54] | South Africa | AFRO | Prospective study | 53 | 47.1 | N = 41, 77.3% | CCA, CB, ICA, both sides, far wall | Mean and maximum CIMT | Reported | DICOM- Not used, Transducer- Linear:11MHz, Edge detection: Not used | Not used | Age |
| Nonterah et al. 2018 [55] | Sub-Saharan Africa | AFRO | Cross-sectional study | 8872 | 49.87 | N = 4507, 50.8% | CCA, both sides, far wall | Mean CIMT | Not reported | DICOM- Not used, Transducer- Linear:7.5-10MHz, Edge detection: Not used | Not used | Gender, BMI, TRF, Socio-economic factors and HIV |
| Kamran et al. 2014 [56] | Iran | EMRO | Case control study | 500 | 60 | N = 287, 57.4% | CCA, both sides | Mean and maximum CIMT | Reported | DICOM- Not used, Transducer- Linear:7.5-15MHz, Edge detection: Not used | Not used | Age, gender, Hypertension, smoking, and Hyperlipidemia |
| Pourafkari et al. 2006 [57] | Iran | EMRO | Cross-sectional study | 113 | 44 | NR | CCA and ICA, both sides | Mean CIMT | Not reported | DICOM- Not used, Transducer- Linear:7.5-15MHz, Edge detection: Not used | Not used | Age and gender |
| Mirza et al. 2017 [58] | Pakistan | EMRO | Cross-sectional study | 257 | 45 | N = 97, 38% | CCA and ICA, both sides | Mean CIMT | Not reported | DICOM- Not used, Transducer- Linear: NR, Edge detection: Used | Not used | Age, diabetes, and gender |
| Mustafa et al. 2013 [59] | Sudan | EMRO | Cross-sectional study | 11 | 41.6 | N = 6, 54.5% | CCA, both sides, far wall | Mean CIMT | Not reported | DICOM- Not used, Transducer- Linear:6MHz, Edge detection: Not used | Not used | Age, smoking, and gender |
| Haghi et al. 2005 [60] | Germany | EURO | Case control study | 151 | 61.5 | N = 120, 79.5% | CCA, both sides, far wall. | Mean CIMT | Reported | DICOM- Not used, Transducer- Linear:7.5MHz, Edge detection: Not used | Not used | Age and gender |
| Kotsis et al. 2005 [61] | Greece | EURO | Cross-sectional study | 390 | 61.2 | N = 345, 88.5% | CCA and ICA, both sides, far wall. | Mean CIMT | Not reported | DICOM- Not used, Transducer- Linear:7MHz, Edge detection: Used | Not used | Age, alcoholic, and gender |
| Mauro Amato et al. 2007 [62] | Italy | EURO | Cross-sectional study | 48 | 61 | N = 36, 75% | CCA, CB and ICA, both sides, far and near wall. | Mean CIMT | Not reported | DICOM- Not used, Transducer- Linear:6.7MHz, Edge detection: Not used | Not used | NR |

*(Continued)*

**Table 1.** (Continued)

| Publication | Country | WHO regions | Design | Sample size | Mean age (years) | Male N, % | Carotid segments | IMT definition | Definition of plaque | Ultrasound scan specifications | ECG gating at acquisition | Factors adjusted for |
|---|---|---|---|---|---|---|---|---|---|---|---|---|
| Del Sol et al. 2001 [63] | Netherlands | EURO | Case control study | 1690 | 71 | N = 686,40.6% | CCA, CB and ICA, both sides, far and near wall. | Mean of Max. CIMT | Reported | DICOM- Not used, Transducer- Linear:7.5MHz, Edge detection: Used | Used | NR |
| Ziembicka et al. 2005 [64] | Poland | EURO | Cross-sectional study | 558 | 57.5 | N = 438,78.5% | CCA, CB and ICA, both sides, far and near wall. | Mean of Max. CIMT | Not reported | DICOM- Not used, Transducer- Linear:5-10MHz, Edge detection: Used | Used | Age, gender, hypertension, smoking, alcoholic, FBS diabetes and Obesity |
| Lisowska et al. 2009 [65] | Poland | EURO | Case control study | 231 | 49 | NR | CCA and CB, both sides, far wall. | Mean CIMT | Reported | DICOM- Not used, Transducer- Linear:3-11MHz, Edge detection: Not used | Not used | Age, gender, diabetes, dyslipidemia, and GFR |
| Timo´teo et al. 2013 [66] | Portugal | EURO | Case control study | 300 | 64.5 | N = 176, 58.7% | CCA, both sides, far wall. | Mean of Max. CIMT | Reported | DICOM- Not used, Transducer- Linear:7.5MHz, Edge detection: Not used | Not used | Gender |
| Sait et al. 2003 [67] | Turkey | EURO | Case control study | 233 | 59 | N = 131,56.2% | CCA, both sides, far wall. | Mean CIMT | Reported | DICOM- Not used, Transducer- Linear:7.5MHz, Edge detection: Not used | Not used | Age, SBP, smoking, alcoholic, diabetes mellitus and total cholesterol |
| SelcanKoc et al. 2019 [68] | Turkey | EURO | Retrospective study | 644 | 54.6 | N = 314,48.5% | CCA and ICA, both sides, far wall. | Mean of Max. CIMT | Reported | DICOM- Not used, Transducer- Linear:5-12MHz, Edge detection: Used | Not used | Age. gender and SBP, FBS |
| Mehmet et al. 2006 [69] | Turkey | EURO | Case control study | 144 | 53.2 | N = 87, 60.4% | CCA and CB, both sides, far wall. | Mean of Max. CIMT | Reported | DICOM- Not used, Transducer- Linear:NR, Edge detection: Not used | Not used | NR |
| Geroulakos et al. 1994 [70] | UK | EURO | Case control study | 122 | 58 | NR | CCA, both sides, far wall. | Mean CIMT | Not reported | DICOM- Not used, Transducer- Linear:7.5MHz, Edge detection: Not used | Not used | NR |
| Ebrahim et al. 1999 [71] | UK | EURO | Cross-sectional study | 800 | 66 | N = 425,53.1% | CCA and CB, both sides, far wall. | Mean of Max. CIMT | Reported | DICOM- Not used, Transducer- Linear:7MHz, Edge detection: Used | Used | Age, gender, Alcohol, smoking, BMI, hypertension, FBS and social class |
| Alejandro et al. 2018 [72] | Argentina | PAHO | Cross-sectional study | 1012 | 42 | N = 621, 61.36% | CCA, ICA and ECA, both sides, far wall | Mean CIMT | Reported | DICOM- Not used, Transducer- Linear:4-13MHz, Edge detection: Used | Used | Gender, FBS, SBP, MBP, DBP, and PP |
| Rosa et al. 2003 [73] | Brazil | PAHO | Case control study | 58 | 50.1 | N = 32,55.2% | CCA: both sides, far walls | Mean CIMT | Not reported | DICOM- Not used, Transducer- Linear:5MHz, Edge detection: Used | Not used | Alcoholic, Smoking, dyslipidemia |
| Amer et al. 2016 [74] | Canada | PAHO | Case control study | 318 | 64 | N = 128, 40.3% | CCA, CB, ICA, both sides, far and near wall | Mean CIMT | Reported | DICOM- Used, Transducer- Linear:7.5MHz, Edge detection: Used | Used | Age |
| Catherine et al. 2010 [75] | USA | PAHO | Cross-sectional study | 472 | 52.4 | N = 214,45.3% | CCA Both sides, Far wall | Mean CIMT | Reported | DICOM- Not used, Transducer- Linear:NR, Edge detection: Used | Used | Age, gender, FBS, diabetes mellitus, dyslipidemia, and smoking |

*(Continued)*

**Table 1.** (Continued)

| Publication | Country | WHO regions | Design | Sample size | Mean age (years) | Male N, % | Carotid segments | IMT definition | Definition of plaque | Ultrasound scan specifications | ECG gating at acquisition | Factors adjusted for |
|---|---|---|---|---|---|---|---|---|---|---|---|---|
| Polak et al. 2011 [76] | USA | PAHO | Case control study | 2965 | 60.1 | N = 1336, 45.1% | CCA, both sides, far wall | Mean CIMT | Reported | DICOM- Not used, Transducer-Linear:12MHz, Edge detection: Used | Used | NR |
| Cao et al. 2007 [77] | USA | PAHO | Cross-sectional study | 5020 | 72.6 | N = 2008,40% | CCA and ICA: near and far walls on both sides. | Mean and Maximum CIMT | Reported | DICOM- Not used, Transducer- Linear:7-10MHz, Edge detection: Not used | Not used | Age, gender, CRP levels |
| Chambless et al. 1997 [78] | USA | PAHO | Case control study | 12841 | 55.3 | N = 5552,43.2% | CCA, CB, ICA, both sides, far wall | Mean CIMT | Not reported | DICOM- Not used, Transducer- Linear:NR, Edge detection: Not used | Not used | Age, race, gender FBS, diabetes, LDL, HDL, hypertension, smoking status |
| Hensley et al. 2020 [79] | USA | PAHO | Case control study | 58 | 60 | N = 39,67.2% | CCA: both sides, far walls | Mean and Maximum CIMT | Reported | DICOM- Not used, Transducer-Linear:7.5MHz, Edge detection: Used | Not used | NR |
| Gupta et al. 2003 [80] | India | SEAR | Case control study | 241 | 47.2 | N = 205, 85.1% | CCA, CB and ICA, both sides, far wall | Mean and Maximum CIMT | Reported | DICOM- Not used, Transducer-Linear:7.5MHz, Edge detection: Not used | Used | Age and gender |
| Sudhir et al. 2018 [81] | India | SEAR | Case control study | 200 | 43.1 | NR | CCA and ICA, both sides | Mean CIMT | Not reported | DICOM- Not used, Transducer- Linear:5-12MHz, Edge detection: Not used | Not used | Age |
| Agarwal et al. 2008 [82] | India | SEAR | Case control study | 111 | 59.2 | N = 66, 59.4% | CCA, both sides, far wall | Mean CIMT | Reported | DICOM- Not used, Transducer-Linear:7.5MHz, Edge detection: Not used | Not used | NR |
| Kasliwal et al. 2016 [83] | India | SEAR | Cross-sectional study | 818 | 43 | N = 438, 53.5% | CCA, both sides, far wall | Mean CIMT | Reported | DICOM- Used, Transducer-Linear:7.5MHz, Edge detection: Used | Used | Age, SBP, FBS, BMI, DBP and serum triglycerides |
| Paul et al. 2012 [15] | India and Bangladesh | SEAR | Cross-sectional study | 96 | 44.34 | N = 53,55.2% | CCA and ICA, both sides, far wall | Mean CIMT | Reported | DICOM- Not used, Transducer- Linear:7.5MHz, Edge detection: Used | Used | Age and gender |
| Rinambaan et al. 2016 [84] | Indonesia | SEAR | Cross-sectional study | 356 | 56 | N = 236, 66.3% | CCA, both sides, Near and far wall | Mean and Maximum CIMT | Reported | DICOM- Not used, Transducer- Linear:7.5-10MHz, Edge detection: Not used | Not used | Age, triglyceride levels had association. But Weight, BMI, Waist circumference, Glucose, LDL-c, HDL-c. |
| Barakoti et al. 2016 [85] | Nepal | SEAR | Case control study | 104 | 55.1 | N = 59, 56.7% | CCA, both sides, far wall | Mean CIMT | Not reported | DICOM- Not used, Transducer-Linear:10MHz, Edge detection: Not used | Not used | NR |
| Adams et al. 1995 [86] | Australia | WPR | Cross-sectional study | 350 | 60 | N = 249,71% | CCA, both sides, far wall. | Mean and Maximum CIMT | Reported | DICOM- Not used, Transducer-Linear:7MHz, Edge detection: Not used | Used | NR |
| Bin Liu et al. 2017 [87] | China | WPRO | Cross-sectional study | 3789 | 58.8 | N = 1560,41.2% | CCA, Both sides, far and near wall. | Mean CIMT | Not reported | DICOM- Not used, Transducer- Linear:5-12MHz, Edge detection: Used | Not used | Age, gender, low education level, smoking, hypertension, SBP, FBS and LDL-c |

*(Continued)*

**Table 1.** (Continued)

| Publication | Country | WHO regions | Design | Sample size | Mean age (years) | Male N, % | Carotid segments | IMT definition | Definition of plaque | Ultrasound scan specifications | ECG gating at acquisition | Factors adjusted for |
|---|---|---|---|---|---|---|---|---|---|---|---|---|
| Xuefang et al. 2020 [88] | China | WPR | Cross-sectional study | 1039 | 72.3 | N = 498,47.9% | CCA, both sides, far and near wall. | Mean CIMT | Reported | DICOM-Used, Transducer- Linear:5-12MHz, Edge detection: Used | Not used | Age, gender and hypertension, FBS |
| Fujihara et al. 2014 [89] | Japan | WPR | Case control study | 116 | 60.5 | N = 78,67.2% | CCA, both sides, far wall. | Mean and Maximum CIMT | Reported | DICOM- Not used, Transducer- Linear:7.5MHz, Edge detection: Used | Not used | NR |
| Matsushima et al. 2004 [90] | Japan | WPR | Case control study | 103 | 62 | N = 71, 68.9% | CCA not mentioned sides and wall | Mean CIMT | Not reported | DICOM- Used, Transducer- Linear:7.5MHz, Edge detection: Used | Not used | Age, BMI, SBP, DBP, HDL-c, LDL-c and HbA1C |
| Young-Hoon et al. 2014 [91] | Korea | WPR | Cross-sectional study | 2595 | 58.7 | N = 713,27.5% | CCA and CB, both sides far wall. | Mean CIMT | Reported | DICOM- Not used, Transducer- Linear:7.5 MHz, Edge detection: Used | Not used | Age, Metabolic syndrome |
| Young Jin et al. 2011 [92] | Korea | WPR | Cross-sectional study | 433 | 55 | N = 107,24.7% | CCA, both sides, far wall | Mean CIMT | Not reported | DICOM- Not used, Transducer- Linear:NR, Edge detection: Not used | Used | Age, gender, BMI, LDL-C level and history of diabetes mellitus. |
| Chua et al. 2014 [93] | Malaysia | WPR | Cross-sectional study | 123 | 55 | N = 74,60.2% | CCA, both sides, far and near wall. | Mean and Maximum CIMT | Not reported | DICOM- Not used, Transducer- Linear:13MHz, Edge detection: Used | Not used | Age, TC and LDL-c |
| Ta-Chen et al. 2015 [94] | Taiwan | WPR | Cross-sectional study | 689 | 51 | N = 497,72.1% | CCA, CB, ICA, both sides, far wall | Mean and Maximum CIMT | Reported | DICOM- Used, Transducer- Linear:3.5-10MHz, Edge detection: Used | Used | Age, gender, diabetes and air pollution |

AFR: African Region, EMRO: Eastern Mediterranean Region, EUR: European Region, PAHO: Region of the Americas, SEAR: South-East Asia Region, WPR: Western Pacific Region, CCA: Common carotid artery, CB: Carotid bulb, ICA: internal carotid artery, ECA: External carotid artery, IMT: Intima-media thickness, SBP: Systolic blood pressure, DBP: Diastolic blood pressure, PP: Pulse pressure, FBS: Fasting blood sugar, TC: Total cholesterol, LDL-c: Low-density lipoprotein cholesterol, HDL-c: High-density lipoprotein cholesterol, HbA1C:, CRP: C-reactive protein, GFR: Glomerular filtration rate, TRF: Traditional risk factors, BMI: Body mass index, HIV: human immunodeficiency virus, DICOM: Digital Imaging and Communications in Medicine, NR: Not reported.

**Table 2. The mean CIMT values of different carotid segments by WHO region and CHD group.**

| Segment | CCA | | | | CB | | | | ICA | | | |
|---|---|---|---|---|---|---|---|---|---|---|---|---|
| Group | Non-CHD | | CHD | | Non-CHD | | CHD | | Non-CHD | | CHD | |
| WHO region | N | Mean±SD (mm) | N | Mean±SD (mm) | N | Mean±SD (mm) | N | Mean±SD (mm) | N | Mean±SD (mm) | N | Mean±SD (mm) |
| AFR | 9244 | 0.70±0.08[a] | 4380 | 0.92±0.13* | 8994 | 0.75±0.06[a] | | NR | 8994 | 0.69±0.06[a] | | NR |
| EMR | 870 | 0.58±0.09[b] | 261 | 0.86±0.26** | 500 | 0.71±0.12[b] | | NR | | NR | | NR |
| EUR | 4668 | 0.74±0.11[c] | 698 | 0.92±0.20* | 877 | 0.77±0.11[c] | 145 | 0.93±0.19* | 464 | 0.66±0.11[b] | 255 | 0.86±0.16 |
| PAHO | 22628 | 0.71±0.07[d] | 839 | 0.89±0.15*** | 8457 | 0.81±0.09[d] | 1798 | 0.93±0.16* | 1802 | 0.65±0.10[b] | 720 | 0.87±0.17 |
| SEAR | 1004 | 0.62±0.10[e] | 251 | 0.87±0.21** | | NR | | NR | | NR | | NR |
| WPR | 8215 | 0.61±0.06[e] | 1391 | 0.87±0.16** | 239 | 0.65±0.08[e] | 239 | 0.89±0.19** | | NR | | NR |

AFR: African Region, EMR: Eastern Mediterranean Region, EUR: European Region, PAHO: Region of the Americas, SEAR: South-East Asia Region, WPR: Western Pacific Region, CCA: Common carotid artery, CB: Carotid bulb, ICA: internal carotid artery, NR: Not reported. N: Number of participants.

Note: The pooled mean was calculated weighting the studies on sample size.

[a,b,c,d,e]Means having a superscript with the same letter are similar (Non-CHD group).

*, **, *** Means having a superscript with the same letter are similar (CHD group).

WPR countries were significantly different from those of countries from AFR, EMR, EUR and PAHO regions (P<0.01). There were significant differences in the mean CIMT values of CB between the regions (P<0.01). The mean CIMT value of ICA was significantly higher in countries in AFR in comparison to countries EUR and PAHO (P<0.01).

The mean CIMT values of CCA vary from 0.86±0.26mm to 0.92±0.20mm. The mean CIMT of CB ranges from 0.89±0.19mm to 0.93±0.19mm. The range for the mean CIMT of ICA was 0.86±0.16mm to 0.87±0.17. In each region, the highest mean CIMT value was in the CB. The highest CIMT values of CCA were reported in AFR and EUR countries. The mean CIMT values of CCA in EMR, SEAR and WPR countries were significantly different from those of countries from AFR, EUR and PAHO regions (P<0.01). There were significant differences in the mean CIMT values of CB in WPR countries in comparison to EUR and PAHO countries (P<0.01). The mean CIMT value of ICA was not significantly different in EUR and PAHO (t-test: 0.819; df: 973; p = 0.793) (Table 2).

## Meta-analysis

The pooled mean CIMT value for healthy persons in all regions was 0.65mm (95%CI–0.62–0.69; $I^2$ = 13.79%) (Fig 3). There was a significant difference in the mean CIMT values between the regions (Test of group difference, $Q_{(40)}$ = 11.51, P = 0.04). Subgroup analyses show no significant difference of mean CIMT values within the regions. Countries in AFR, (0.72mm), PAHO (0.71mm) and EUR (0.71mm) had a higher pooled mean CIMT compared to countries in SEAR (0.62mm), WPR (0.60mm) and EMR (0.60mm) (Fig 4). The pooled mean CIMT values were significantly different between different age groups ($Q_{(3)}$ = 19.32, P<0.001) (Fig 4).

The pooled mean CIMT difference between healthy males and females was 0.06mm (95% CI: 0.04–0.07). There were differences in the mean CIMT between males and females within regions (AFR: 0.04mm, p = 0.04; PAHO: 0.05mm, p<0.001: and WPR: 0.04mmp<0.001) (Fig 5).

There was a significant mean difference of the pooled CIMT values between CHD and non-CHD groups (0.23mm, p = 0.001) (Fig 6). PAHO ($I^2$ = 97.18%, $Q_{(3)}$ = 56.63, p<0.001), SEAR ($I^2$ = 99.22%, $Q_{(3)}$ = 376.54, p<0.001) and EUR ($I^2$ = 78.98%%, $Q_{(4)}$ = 18.13, p<0.001)

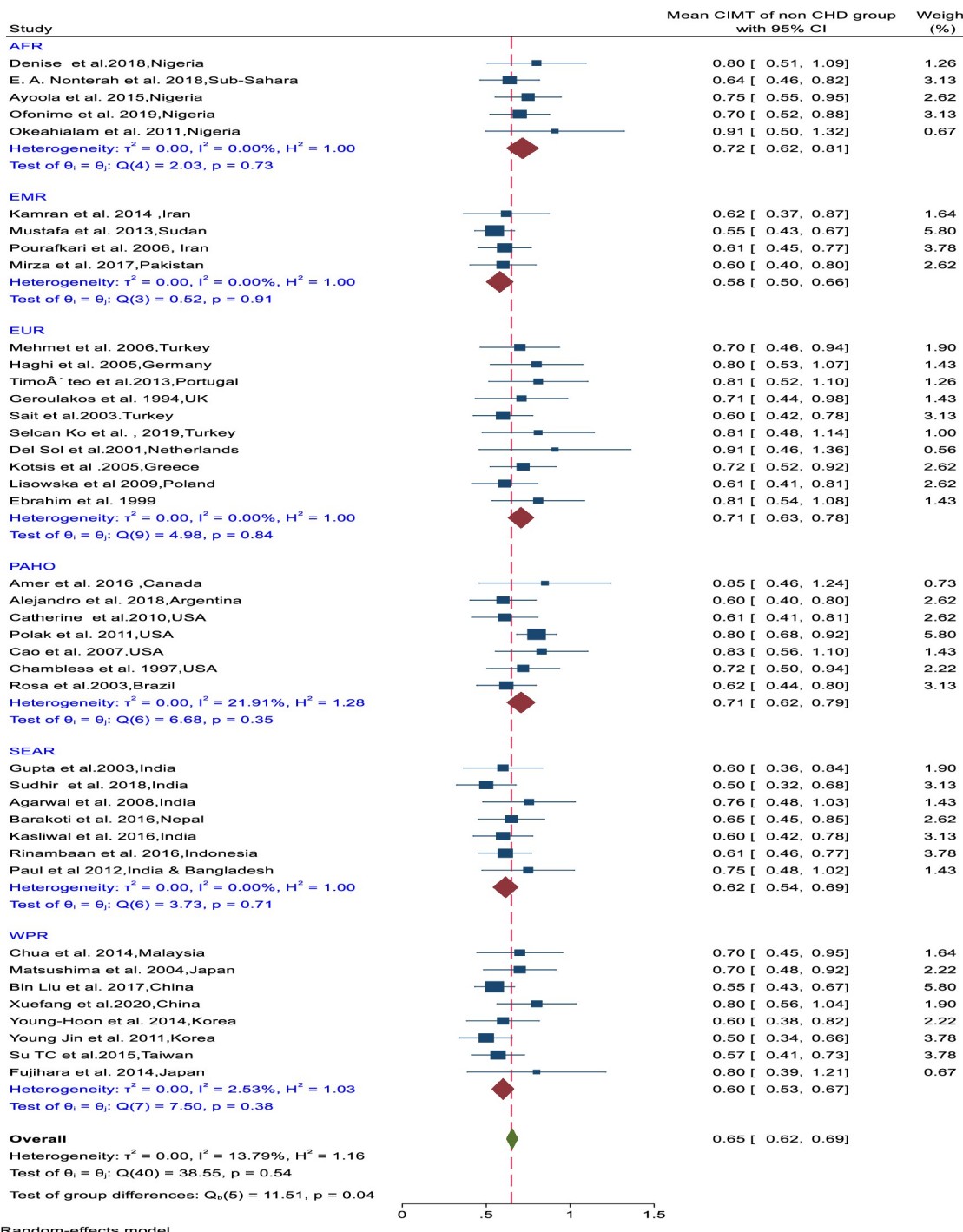

**Fig 3. Forest plot of the mean CIMT for healthy persons by WHO regions.**

countries had significant differences in the mean CIMT difference between the CHD and non-CHD groups within the respective region (Fig 6).

Table 3 shows the summary of the Meta regression analysis of CIMT values. In the adjusted model, CHD group, WHO region and age were significantly associated with CIMT. The mean

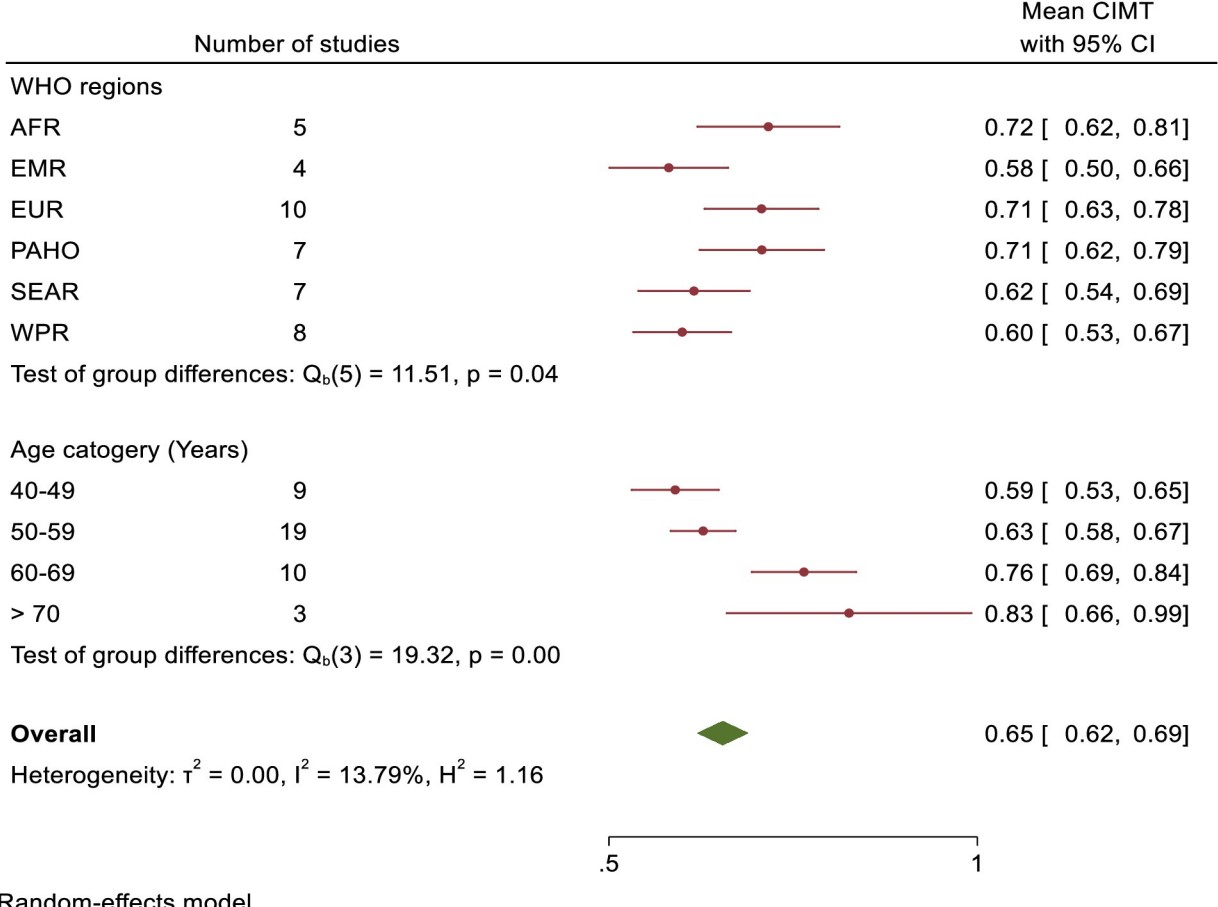

**Fig 4. Summary of mean CIMT values by age and WHO region in healthy persons.**

CIMT value in the CHD group was 0.214 mm greater than that of the non-CHD group after adjusting for the other variables. The mean CIMT was significantly less among populations in SEAR and WPR as compared to populations from PAHO after adjustment. With age there was a significant increase in the mean CIMT values.

## Discussion

Coronary heart disease (CHD) is the most important cause of morbidity, mortality and premature deaths of NCDs. We included 46 eligible articles comprising data of 49 381 individuals. The highest number of studies was from the European region while the lowest was from the Eastern Mediterranean region.

### Modifiable risk factors

45.6% of the studies reviewed showed that modifiable risk factors were predictors of CIMT. There was a significant difference in CIMT values among the non-CHD group between regions. Higher CIMT values were observed in countries in the African, American and European regions. The mean difference in CIMT values between CHD and non-CHD groups were significantly different between and within regions. Differences in the CIMT values between regions may be due to socio-economic status [95, 96], environmental conditions, smoking

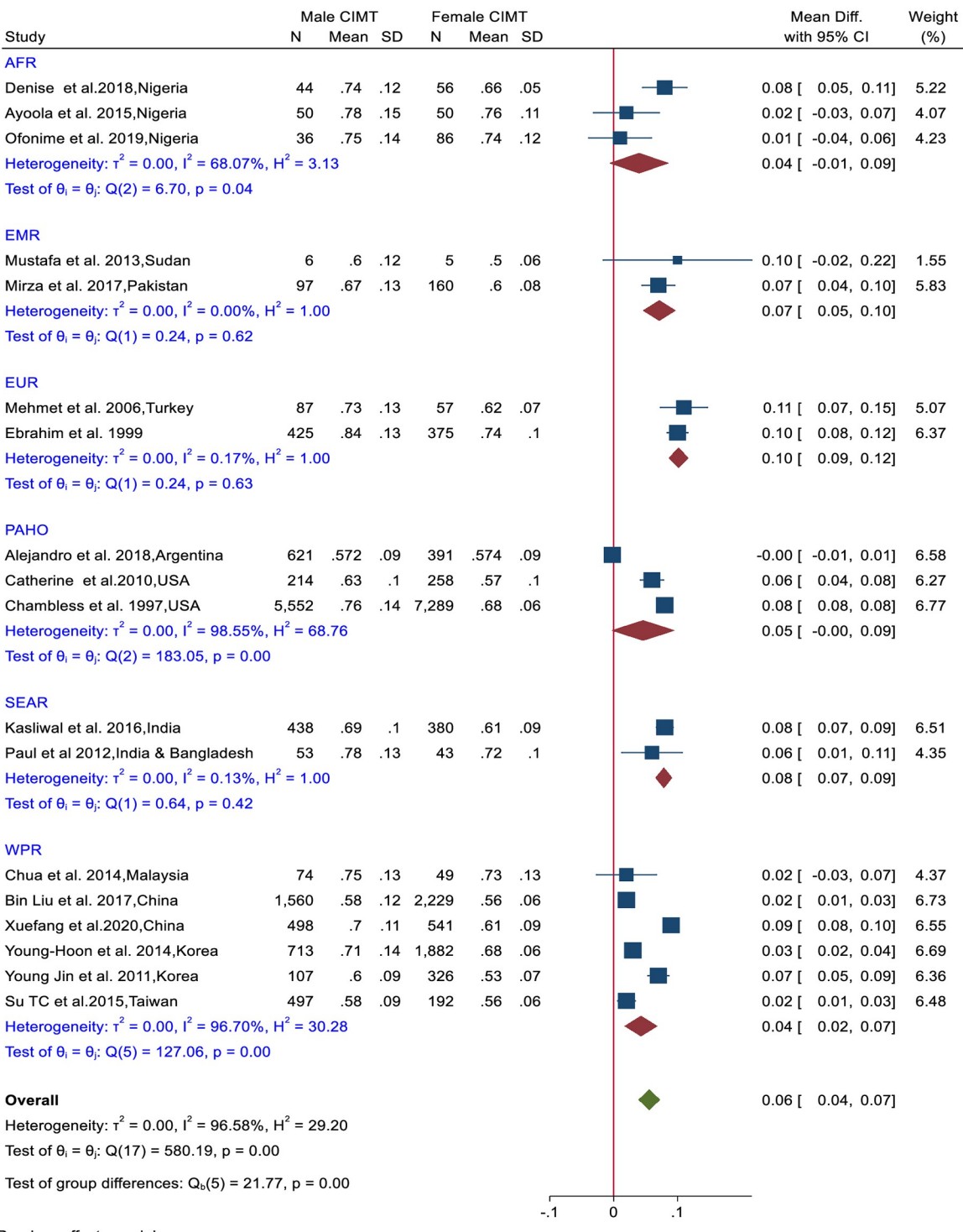

**Fig 5. Forest plot of mean difference of CIMT between healthy males and females.**

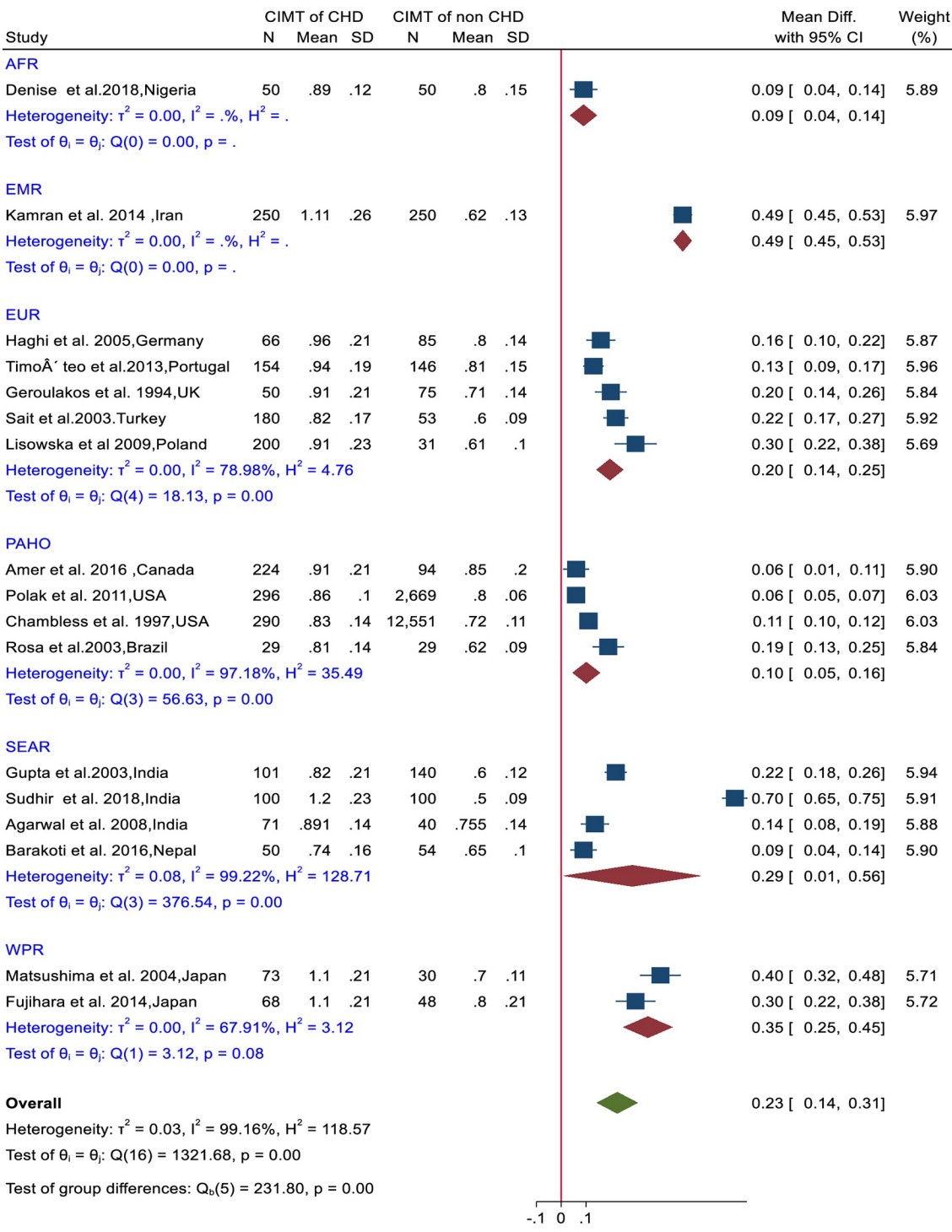

**Fig 6. Forest plot of mean difference of CIMT between CHD and non-CHD groups.**

**Table 3. Summary of meta regression analysis of CIMT values.**

| Variable | Unadjusted Regression coefficient | 95%CI | Adjusted regression coefficient | 95%CI |
|---|---|---|---|---|
| Age (years) | 0.008* | 0.004 to 0.013 | 0.006* | 0.001 to 0.011 |
| AFR | 0.001 | -0.121 to 0.122 | 0.026 | -0.112 to 0.175 |
| EMR | -0.141* | -0.256 to -0.027 | -0.064 | -0.217 to 0.087 |
| EUR | 0.006 | -0.096 to 0.109 | -0.013 | -0.141 to 0.112 |
| SEAR | -0.173* | -0.279 to -0.067 | -0.149* | -0.287 to -0.012 |
| WPR | -0.107* | -0.207 to -0.006 | -0.117* | -0.217 to -0.165 |
| Region of the Americas | Reference category | | | |
| Automatically | -0.050 | -0.144 to 0.044 | 0.016 | -0.067 to 0.099 |
| Automatically with ECG gating | -0.023 | -0.142 to 0.094 | 0.018 | -0.097 to 0.133 |
| Manual ultrasound technique | Reference category | | | |
| CHD group | 0.228* | 0.153 to 0.304 | 0.214* | 0.139 to 0.289 |
| Non CHD group | Reference category | | | |
| Constant | | | 0.578 | |

AFR: African Region, EMR: Eastern Mediterranean Region, EUR: European Region, PAHO: Region of the Americas, SEAR: South-East Asia Region, WPR: Western Pacific Region.

*significant variables.

habits, harmful consumption of alcohol, physical activity, dietary patterns, sedentary behaviors and body mass indices [23, 24, 97–100], and prevalence of co-morbidities such as diabetes, hypertension, dyslipidemia, cancer and chronic kidney disease [101, 102].

Age-adjusted cardiovascular death rates have declined in several developed countries in the past decades. In contrast, the death rates of cardiovascular disease have risen greatly in lower middle income countries [103–105]. Several publications underscore the high burden of disease associated with non-communicable diseases and its economic impact on lower middle income countries [4, 104, 106]. Due to this reason, non-communicable diseases in lower middle income countries have received increasingly more global attention by scientists, public health advocates and policy makers. A recent study has identified that NCDs and CHD risk factors such as demographic transition, environmental pollution, metabolic risk factors, lack of education, unhealthy food habits and unhealthy lifestyles have similar effects in both developed and developing countries [109].

Some studies reported that non-traditional risk factors such as HIV infection, metabolic syndrome, infections and inflammation as predictors of CIMT. Some studies have highlighted that during chronic infections and inflammation, elevated levels of the pro-inflammatory cytokines interleukin (IL)-6 and C-reactive protein (CRP) are associated with subclinical atherosclerosis [107, 108]. Intima-medial thickening is a complex process. Modifiable risk factors contribute in different stages in different proportions. Factors that vary stress and blood pressure, which may cause a local delay in lumen transportation, may lead to the accumulation of potentially atherogenic particles in the arterial wall and stimulate CIM thickening and plaque formation [109]. Risk factors which cause endothelial destruction and functional abnormalities are associated with higher carotid IMT and were associated with a higher risk of atherosclerotic disease [110].

## Non- modifiable risk factors

28.5% of the studies we reviewed reported that non-modifiable risk factors such as age and gender are associated with CIMT. The CIMT values of males are significantly higher than that

of females (pooled difference of 0.06 mm) across regions. In our meta-analysis there was a significant difference in the pooled mean CIMT values between the age groups with older age groups having higher CIMT values.

Heredity and certain genotypes [27, 28], immunological diseases [111, 112], inflammatory cytokines, hematological parameters [30,112–114] and vitamin D [115] have been reported to be potential risk factors for increased CIMTs. In our review, we did not find these to be risk factors probably due to the specific study designs, study populations and outcomes considered by us.

The Meta regression analysis demonstrated that CIMT values were influenced by WHO region, age and CHD group. Even though there is a clear association between CIMT and CHD its usability as a risk predictor for CHD needs to be further investigated. Approaches to prevention as well as screening of at-risk populations for CHD may need to consider regional variations of CIMT.

Most studies included in this review had not documented the ethnic composition of their samples. Therefore, we were unable to evaluate CIMT variations among different ethnicities. It is reported that healthy UK black African-Caribbean children have higher CIMT levels, not explained by conventional cardiovascular risk markers, as compared to other ethnicities [116]. Ethnicity significantly modifies the associations between risk factors, CIMT values and cardiovascular events [122]; the association between CIMT and age, HDL cholesterol, total cholesterol and smoking was weaker among Blacks and Hispanics [117]. Systolic blood pressure was associated more strongly with CIMT in Asians [117]. These differences could be due to varying interactions between different risk factors and ethnicities. These differences provide insight into the etiology of cardiovascular disease among ethnic groups and aid the ethnic-specific implementation of primary prevention.

## Segmental variation of CIMT

We have summarized variations in the mean CIMT values of CCA, CB and ICA within and between regions. These differences may be due to different influences of risk factors on the different segments. A Korean study reported associations between cardiovascular risk factors and different segments of the carotid artery: in men, alcohol use (CIMT at the bifurcation); physical activity (CIMT at the common and internal carotid segments); BMI (CIMT of all segments); diabetes (CIMT at the bifurcation and internal carotid segment); hypertension (CIMT at the internal carotid segment); and HDL-cholesterol (CIMT at the bifurcation and the common carotid segment): in women, smoking (CIMT at the bifurcation), hypertension (CIMT at the common carotid segment), total and LDL cholesterol (CIMT at the bifurcation and internal carotid segment), and hs-CRP (CIMT at the common and internal carotid segments) [118]. Furthermore, the Malmö Diet and Cancer Study (MDCS) reported that HDL was associated with IMT progression in the CCA but not at the bifurcation. The same showed that diabetes was associated with IMT progression at the bifurcation, but not in the CCA [119, 120].

This study summarized that CIMT values of non-CHD population vary among regions. Age and gender have a significant effect on CIMT differences. Furthermore, there were marked differences of mean CIMT values between non-CHD and CHD groups. It was different from region to region as well as within regions.

## Ultrasound protocol for CIMT measurement

There were variations in the ultrasound assessment of CIMT. The transducer frequency ranged from 3MHz to 15MHz; five studies used DICOM software. Variations in the ultrasonography protocol are likely to affect CIMT values. There are different arguments with regard to various ultrasound protocols during CIMT measurement [44]. Mannheim Carotid Intima-

Media Thickness consensus (2004–2006) is a useful guideline to achieve homogeneity of CIMT measurement among studies [121]. A common protocol will ensure reproducibility and comparison of findings of different studies.

There was no uniformity in the selection of the site for measurement or the reporting of the CIMT measurement. The far wall of CCA (both sides) was the commonest site (41%) selected. The mean CIMT value was reported in 65% of studies. Plaque formation was reported in 59% of studies. It has been reported that this is unlikely to alter the results by much in populations with a low prevalence of plaque [44]. Some studies imaged only one side of the neck, whereas others imaged both sides [122]. Some included imaging of a single segment while multiple segments were imaged in others [77, 123, 124]. Some studies imaged the far wall of multiple segments, whereas others imaged both the near and far walls [125, 126]. Studies also differed in the type of IMT measurements made and the use of different arbitrary cut-off points of CIMT to predict risk. Our review also shows that ECG gating at acquisition was reported only by 28% of studies. The phase of the cardiac cycle (end-systole vs. end-diastole) when CIMT is measured also differs among studies. Because of systolic lumen diameter expansion that leads to thinning of CIMT during systole, CIMT values obtained from end-systole are lower than those obtained in end-diastole [16]. In our meta-analysis we categorized ultrasound technique of measuring CIMT into three categories; manually, automatically and automatically with ECG gating. Literature shows that CIMT measured by General Electric (GE) semi-automated edge-detection software and Artery Measurement semi-automated software (AMS) have significant differences when measuring mean CIMT [127]. Hence, results obtained from different CIMT software systems should be compared with caution. CIMT variations using similar software may be explained by the position/angle of ultrasound transducer, and the specific combinations of segments and walls examined [128]. These factors are associated with differences in reproducibility, magnitude, and precision of progression of CIMT over time. To avoid these discrepancies, it is recommended to measure CIMT in multiple segments with different angles [128]. In our review, we found that most of the studies have obtained an average CIMT value by multiple measurements. This may be a reason that significant differences were not found when multiple segments were examined.

## Conclusion

CIMT among the non-CHD group varies between and within regions, and by age and sex. The mean CIMT values between non-CHD and CHD groups were significantly different within and between WHO regions possibly due to varying influences of modifiable and non-modifiable risk factors. CHD group had a significantly thicker mean CIMT after adjusting for age, WHO region and ultrasound machine used. Segment specific CIMT variations exist among regions.

## Limitation of study

Our review consisted of few studies with small sample sizes. But Egger's test showed no significant small study effect in our review. Some studies had large sample sizes which contributed most to our analyses. We were unable to capture some new risk factors such as genetic composition, immune disorders and cytokine's effect on CIMT due to the selection criteria we used. However, these studies had small sample sizes which may not have been generalizable. It is unlikely that exclusion of these risk factors would have influenced our findings.

The way we grouped countries by WHO regions may not be the most appropriate grouping to consider as WHO regions have been established taking into consideration political considerations as well. For example, the Republic of Korea (South Korea) is in the WPR whereas the Democratic People's Republic of Korea (North Korea) is in the SEAR. Similarly, Pakistan and

Afghanistan, both South Asian countries, are in the EMR, though all other South Asian countries (Bangladesh, Bhutan, India, Maldives, Nepal and Sri Lanka) are in the SEAR together with Myanmar, Thailand, Indonesia and Timor Leste. Therefore, there is a likelihood of ethnic and cultural diversity influencing CIMT values among countries within regions.

Studies included in our review had not specified the ethnic composition of the study samples. Consequently, we were unable to examine CIMT variations by ethnicity. Further studies to explore this variability in future are warranted.

## Supporting information

**S1 File. Search strategy for PubMed.**
(DOCX)

**S1 Table. Summary of STROBE statement.**
(XLSX)

**S2 Table. Newcastle–Ottawa Scale for cross sectional study.**
(XLSX)

**S3 Table. Newcastle–Ottawa Scale for case—Control study.**
(XLSX)

**S4 Table. Newcastle–Ottawa Scale for Cohort studies.**
(XLSX)

**S1 Checklist. PRISMA checklist.**
(DOCX)

## Acknowledgments

The authors thank Department of Public Health, Faculty of Medicine, University of Kelaniya for providing support and resources.

## Author Contributions

**Conceptualization:** V. Abeysuriya, B. P. R. Perera, A. R. Wickremasinghe.

**Data curation:** V. Abeysuriya.

**Formal analysis:** V. Abeysuriya, B. P. R. Perera, A. R. Wickremasinghe.

**Funding acquisition:** V. Abeysuriya.

**Investigation:** V. Abeysuriya.

**Methodology:** V. Abeysuriya, B. P. R. Perera, A. R. Wickremasinghe.

**Project administration:** V. Abeysuriya, A. R. Wickremasinghe.

**Resources:** V. Abeysuriya, B. P. R. Perera.

**Software:** V. Abeysuriya.

**Supervision:** A. R. Wickremasinghe.

**Validation:** V. Abeysuriya, B. P. R. Perera.

**Writing – original draft:** V. Abeysuriya, A. R. Wickremasinghe.

**Writing – review & editing:** V. Abeysuriya, B. P. R. Perera, A. R. Wickremasinghe.

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
