## [Decision Letter · Decision Letter 0]

10 Feb 2022

PONE-D-21-23252

Regional and demographic variations of Carotid artery Intima and Media Thickness (CIMT): A systemic review and meta-analysis.

PLOS ONE

Dear Dr. Abeysuriya,

Thank you for submitting your manuscript to PLOS ONE. After careful consideration, we feel that it has merit but does not fully meet PLOS ONE’s publication criteria as it currently stands. Therefore, we invite you to submit a revised version of the manuscript that addresses the points raised during the review process.

We look forward to receiving your revised manuscript.

Kind regards,

Andreas Zirlik, MD

Academic Editor

PLOS ONE

Journal Requirements:

2. Thank you for stating the following financial disclosure: "VA received the funds from Nawaloka Hospital Research and Education Foundation, Nawaloka hospitals PLC, Colombo-02 Sri Lanka. Grant number is NHREF/01/2020.Funder web site:https://www.nawaloka.com.The funders had no role in study design, data collection and analysis, decision to publish, or preparation of the manuscript."

We note that one or more of the authors is affiliated with the funding organization, indicating the funder may have had some role in the design, data collection, analysis or preparation of your manuscript for publication; in other words, the funder played an indirect role through the participation of the co-authors. If the funding organization did not play a role in the study design, data collection and analysis, decision to publish, or preparation of the manuscript and only provided financial support in the form of authors' salaries and/or research materials, please do the following:

a. Review your statements relating to the author contributions, and ensure you have specifically and accurately indicated the role(s) that these authors had in your study. These amendments should be made in the online form.

b. Confirm in your cover letter that you agree with the following statement, and we will change the online submission form on your behalf: 

“The funder provided support in the form of salaries for authors [insert relevant initials], but did not have any additional role in the study design, data collection and analysis, decision to publish, or preparation of the manuscript. The specific roles of these authors are articulated in the ‘author contributions’ section.

Reviewers' comments:

Reviewer's Responses to Questions

**Comments to the Author**

1. Is the manuscript technically sound, and do the data support the conclusions?

Reviewer #1: Partly

Reviewer #2: Partly

2. Has the statistical analysis been performed appropriately and rigorously? 

Reviewer #1: I Don't Know

Reviewer #2: Yes

3. Have the authors made all data underlying the findings in their manuscript fully available?

Reviewer #1: Yes

Reviewer #2: No

4. Is the manuscript presented in an intelligible fashion and written in standard English?

Reviewer #1: Yes

Reviewer #2: Yes

5. Review Comments to the Author

Reviewer #1: In this manuscript Abeysuriya et al. present a meta-analysis on systematic search regional and demographic variations of carotid artery intima and media thickness. In a first step, the authors screened PubMed, Oxford Medicine Online, EBSCO, Taylor & Francis, Oxford University Press and Embase databases (with a supplementary search in Web of Science and Google Scholar) for eligible on carotrid artery intima and media thickness published between January 1980 January up to December 2020. Subsequently, meta-analyses were done using random-effects models.

Of 2847 potential articles, 46 eligible articles were included in the review contributing data for 49 381 individuals. The authors report a significant difference in the mean CIMT between regions, with countries in the African, American and European regions had a higher pooled mean CIMT compared to those in the Southeast Asian, Western Pacific and Eastern Mediterranean regions. Males appeared to have a higher pooled mean CIMT than females in the non-CHD group. The CHD group had a significantly higher mean CIMT than the non-CHD group. Age and region were significant predictors of CIMT among the non-CHD group.

This manuscript presents interesting data, however, several questions remain:

Major comments:

1. Section “Method and analysis”, paragraph “Study selection“: According to this paragraph, titles and abstracts of the search results were screened against pre-specified criteria by two independent reviewers for study selection. Please add a description of the pre-specified criteria for study selection to the text.

2. Section “Method and analysis”, paragraph “Study quality”: The quality of selected studies was assessed using the Quality Assessment of Diagnostic Accuracy Studies (QUADAS-2) criteria. It is important to assess the quality of reports in a consistent manner with other studies. The CONSORT statement and the STROBE statement represent 2 of the most widely accepted and used guidelines for accurate reporting and transparency. Therefore, it is recommended to additionally assess the quality of reports in reference to the CONSORT statement and the STROBE statement prior to including them in the analysis.

3. Section “Method and analysis”, paragraph “Steps of Meta-analyses”: The meta-analyses were performed using random-effects models. Which model was used? Was the model based on inverse variance-weighted average method or weighted sum of z-scores?

4. Section “Method and analysis”, paragraph “Steps of Meta-analyses”: Due to the comparison of highly heterogeneous populations in different WHO regions with very divers individual risk profiles, there is a high chance of confounding. With regard to the extracted CIMT estimates, how did the model of the authors account for the risk of residual confounding? Were the estimates extracted from the primary studies confounder-adjusted or unadjusted data? The extraction process should be clarified. In addition, comparisons of adjusted and crude estimates allow insights into the importance of confounding. To reliably detect independent regional differences as well as potential influencing variables, the random-effects model should be adjusted for all known risk factors for cardiovascular disease, if possible from the data set.

5. Section “Method and analysis”, paragraph “Assessment of heterogeneity of studies”: In addition to the QUADAS-2 criteria, the Newcastle-Ottawa Scale should be applied to minimize risk of bias.

6. Section “Results”: What was the percentage of included subjects from grey literature and from published literature? What proportion of the total number of included patients is derived from each of the databases as a source? These numbers should be added to the results section as well as to the flow diagram in “Figure 1”.

7. Section “Discussion”: The discussion is mainly descriptive with an extended presentation of the results of the meta-analysis and its underlying primary studies. A reflection of possible underlying factors and differential population characteristics of the respective WHO regions would contribute to strengthen the discussion. Furthermore, general text flow and readability of the discussion should be significantly improved.

Minor comments:

1. Section “Abstract”: The search strategy is not clearly described in the abstract. Therefore, adding a summarized description of the search strategy is recommended.

2. Section “Method and analysis”, paragraph “Search strategy”: The search strategy for PubMed should be specified in the text section of the search strategy paragraph. The separate box displaying the paragraph should be removed from the main manuscript.

3. Section “Method and analysis”, paragraph “:Data extraction”: For articles based on the same population, the authors state that the ‘more comprehensive one’ was selected. Please add a precised description of the data extraction criteria.

4. Section “Results”, “Table 3” and “Table 4”: “Table 3” and “Table 4” show mean CIMT values of different carotid segments by WHO region among patients with and without CHD. To provide a more focused illustration of the results, the content of “Table 3” and “Table 4” should be summarized in a single table.

5. Section “Results”: The text flow in the sections results is partially very tough. Therefore, it should be revised to ensure a more fluid presentation of the results.

Reviewer #2: Comments to the authors:

The authors present a systematic review and meta-analysis of differences of carotid intima media thickness (CIMT) in various regions around the globe, based on the WHO definition of regions. Analysis and pooling data of 49 381 patients showed a difference in CIMT between African, American and European population versus Southeast Asian, Western Pacific and Eastern Mediterranean. The authors found significant regional differences of mean CIMT between CHD and non-CHD groups. The authors conclude that CIMT varies according to region, age and sex among the non-CHD group and that there are significant regional differences of mean CIMT between CHD and non-CHD groups. The authors also state that there is a need to develop country-specific CIMT cutoff values to screen at-risk populations for CHD. The following points arose to the reviewers eyes when reading the manuscript:

Major comments:

- Regarding the conclusion, that there is a need to establish country specific CIMT cutoff values, I find it difficult to state this in light of the presented data. For my understanding in this study, differences between countries have not been investigated (since the results are based on WHO regions). As the authors also state in the discussion, there are many factors that affect CIMT, and there are known regional differences in the prevalence of those risk factors (e.g. BMI, hypertension, diabetes). Since reference values are based on studies on a healthy population, it would be interesting to look at geographical differences in comparing healthy cohorts. Is there a possibility to address this? Otherwise I would recommend to rewrite this part of the conclusion.

- Ethnical aspects: It is known that in regards to cardio- and cerebrovascular disease the ethnical background plays a detrimental role. The presented study seems to calculate the values of patients from countries, but irrespective of their ethnicity. Is there also a possibility to analyze the impact of the ethnical background?

- Geographical aspects: There is also a different burden of cardiovascular and cerebrovascular diseases within one WHO region, e.g. Northern Europe versus southern Europe. Some regions also seem to be underrepresented. Authors tried to apply statistical methods relativize this fact and they also comment this in the study limitations section, that WHO classification is also partly based on political aspects as well. Would it not be clinically more reasonable to compare trials of comparable quality using a classification that is only based on geographical aspects (even taking into account not to cover the whole globe)?

- Time span: Authors have reviewed and compared data on CIMT in the time span from 1980 to 2020. Within 40 years, the spatial resolution of ultrasound systems has revolutionized and is still becoming more precise, and therefore the measurements of vessels and their segments are not entirely comparable. Furthermore, there are significant differences in measuring the CIMT manually and automatically. Further factors that have changed over the decades are methodical quality of trials, the quality of trial performance, and the presence of trial audits are not respected in the study and are very likely to influence the outcome. Would it be possible to analyze only studies with a similar technological standard? This would improve the value of this study, since it would minimize bias due to technical and methodical differences.

- The statistical methods seem to be sufficient.

- Search strategy seems to be representative according to selected keywords, but the heterogeneity of population and patients with co-morbidity is not reflected.

Minor Comments:

Line 59: word „diseases“ is missing.

Line 64: The sentence starting with “The current COVID-19 pandemic…” is not relevant to the presented review and I would suggest to delete it.

6. PLOS authors have the option to publish the peer review history of their article (what does this mean?). If published, this will include your full peer review and any attached files.

Reviewer #1: No

Reviewer #2: No

---

## [Author Response · Author response to Decision Letter 0]

19 Mar 2022

1. Academic editor : Many thanks for your valuable comments.We hereby sincerely address the specific academic editor comments and queries. (Please refer to "Response to reviewer" attachment and the revised manuscript: marked-up copy) 

2. Reviewer 1: We would like to thank the reviewer for the comments given in the Review Form of our manuscript.We hereby sincerely address the specific reviewer comments and queries.(Please refer to "Response to reviewer" attachment and the revised manuscript: marked-up copy)

3. Reviewer 2: We would like to thank the reviewer for the comments given in the Review Form of our manuscript.We hereby sincerely address the specific reviewer comments and queries.(Please refer to "Response to reviewer" attachment and the revised manuscript: marked-up copy)

---

## [Decision Letter · Decision Letter 1]

18 Apr 2022

PONE-D-21-23252R1Regional and demographic variations of Carotid artery Intima and Media Thickness (CIMT): A systemic review and meta-analysis.PLOS ONE

Dear Dr. Abeysuriya,

Thank you for submitting your manuscript to PLOS ONE. After careful consideration, we feel that it has merit but does not fully meet PLOS ONE’s publication criteria as it currently stands. Therefore, we invite you to submit a revised version of the manuscript that addresses the points raised during the review process.

We look forward to receiving your revised manuscript.

Kind regards,

Andreas Zirlik, MD

Academic Editor

PLOS ONE

Journal Requirements:

Reviewers' comments:

Reviewer's Responses to Questions

**Comments to the Author**

1. If the authors have adequately addressed your comments raised in a previous round of review and you feel that this manuscript is now acceptable for publication, you may indicate that here to bypass the “Comments to the Author” section, enter your conflict of interest statement in the “Confidential to Editor” section, and submit your "Accept" recommendation.

Reviewer #1: All comments have been addressed

Reviewer #2: (No Response)

2. Is the manuscript technically sound, and do the data support the conclusions?

Reviewer #1: Yes

Reviewer #2: Partly

3. Has the statistical analysis been performed appropriately and rigorously? 

Reviewer #1: Yes

Reviewer #2: Yes

4. Have the authors made all data underlying the findings in their manuscript fully available?

Reviewer #1: (No Response)

Reviewer #2: Yes

5. Is the manuscript presented in an intelligible fashion and written in standard English?

Reviewer #1: Yes

Reviewer #2: Yes

6. Review Comments to the Author

Reviewer #1: Overall the authors were responsive to the previous comments of these reviewers and the manuscript improved subsequently. Crucial aspects were appropriately addressed by the authors’ reply. However in the eyes of this reviewer, some points remain to be clarified:

Major comments?

1. Section “Discussion”: How do the authors interpret the value of these findings? Because of the descriptive background of this analysis and the different adjustment for different risk factors regarding the included studies, it is indeed difficult to demonstrate an incremental value of the participant’s geographical region/country. Therefore, as already indicated by reviewer 2, the topic should be considered very cautiously in the discussion. Statements on causal relationships and incremental value of the variable region for risk prediction should therefore be avoided. Adapt more defensive wording regarding this relationship.

Minor comments:

1. Section “Results”, Table 1: “Summary of studies used in systematic review and meta-analysis, reporting demography: As requested, the authors added further information on the adjustment of the studies included in the analyses. However, the mention of "factors adjusted for" and "adjusted predictors of CIMT" seems repetitive. Therefore, the column "Adjusted predictors of CIMT" should be removed since all relevant information is already listed in the column "Factors adjusted for."

2. Section “Method and analysis”: Grammar and spelling of the newly added text parts should be revised.

3. Section “Results”: Since the included studies and corresponding details are already listed in Table 1, there is no need to cite the respective studies again separately in the results section.

Reviewer #2: Most of the concerns mentioned in the first review of the manuscript were addressed. Nonetheless there are still minor comments regarding the rewritten conclusion of the authors:

The conclusion, that region or country specific CIMT values are important when developing risk assessment tools to screen at-risk population of CHD is not supported by the data presented, since in this study only differences in CIMT between WHO regions were examined and not their impact to a certain CV risk and significant confounding is possible. CIMT in fact may be important in the risk assessment, but the presented data do not fully validate the role of CIMT differences between WHO regions regarding CV risk. Furthermore, in my opinion the conclusion that CHD is a predictor for CIMT is clinically not sound. As for my understanding, the presented data show a clear association between CHD and CIMT and not necessarily that the presence of CHD predicts CIMT values.

Therefore, I would recommend to precise the conclusion according to the presented data which may help to improve the quality of the manuscript.

7. PLOS authors have the option to publish the peer review history of their article (what does this mean?). If published, this will include your full peer review and any attached files.

Reviewer #1: No

Reviewer #2: No

---

## [Author Response · Author response to Decision Letter 1]

28 Apr 2022

Responses to the raised review points by the reviewers

Reviewer 1:

Comment

Major comments:

1. Section “Discussion”: How do the authors interpret the value of these findings? Because of the descriptive background of this analysis and the different adjustment for different risk factors regarding the included studies, it is indeed difficult to demonstrate an incremental value of the participant’s geographical region/country. Therefore, as already indicated by reviewer 2, the topic should be considered very cautiously in the discussion. Statements on causal relationships and incremental value of the variable region for risk prediction should therefore be avoided. Adapt more defensive wording regarding this relationship.

Response

We understand the concerns of the reviewer. We have made the following changes:

Abstract – we reworded and included the following sentences

Older persons and those having CHD group had significantly thicker CIMTs – results section (Line 54) 

There is an association between CHD and CIMT values – conclusion (Line 57)

In the results section of the body of the manuscript the sentence was reworded as

“In the adjusted model, CHD group, WHO region and age were significantly associated with CIMT.” (Line 331)

In the discussion, the following sentence was added.

“Even though there is a clear association between CIMT and CHD its usability as a risk predictor for CHD needs to be further investigated. Approaches to prevention as well as screening of at-risk populations for CHD may need to consider regional variations of CIMT.” (Line 389 to 391)

The following is the conclusion of the manuscript

“CIMT among the non-CHD group varies between and within regions, and by age and sex. The mean CIMT values between non-CHD and CHD groups were significantly different within and between WHO regions possibly due to varying influences of modifiable and non-modifiable risk factors. CHD group had a significantly thicker mean CIMT after adjusting for age, WHO region and ultrasound machine used. Segment specific CIMT variations exist among regions.”(Line 455 to 459)

Minor Comment

1. Section “Results”, Table 1: “Summary of studies used in systematic review and meta-analysis, reporting demography: As requested, the authors added further information on the adjustment of the studies included in the analyses. However, the mention of "factors adjusted for" and "adjusted predictors of CIMT" seems repetitive. Therefore, the column "Adjusted predictors of CIMT" should be removed since all relevant information is already listed in the column "Factors adjusted for."

Response

We removed the column.

Minor Comment

2. Section “Method and analysis”: Grammar and spelling of the newly added text parts should be revised.

Response

Grammar and spelling has been revised.

Minor Comment

3. Section “Results”: Since the included studies and corresponding details are already listed in Table 1, there is no need to cite the respective studies again separately in the results section.

Response

As suggested, correction has been done in the result section by omitting the references. 

Reviewer 2:

Minor comments:

1. The conclusion, that region or country specific CIMT values are important when developing risk assessment tools to screen at-risk population of CHD is not supported by the data presented, since in this study only differences in CIMT between WHO regions were examined and not their impact to a certain CV risk and significant confounding is possible. CIMT in fact may be important in the risk assessment, but the presented data do not fully validate the role of CIMT differences between WHO regions regarding CV risk. Furthermore, in my opinion the conclusion that CHD is a predictor for CIMT is clinically not sound. As for my understanding, the presented data show a clear association between CHD and CIMT and not necessarily that the presence of CHD predicts CIMT values.

Therefore, I would recommend to precise the conclusion according to the presented data which may help to improve the quality of the manuscript.

Response

As suggested by the reviewer, we have deleted the sentence that country-specific CIMT values are important for risk prediction of CHD.

We have reworded the relevant sentences that stated that CHD is a predictor of CIMT. Please see response to major comment by reviewer 1 above which gives all details of the corrections made.

---

## [Editor Report · Decision Letter 2]

6 May 2022

Regional and demographic variations of Carotid artery Intima and Media Thickness (CIMT): A systemic review and meta-analysis.

PONE-D-21-23252R2

Dear Dr. Abeysuriya,

We’re pleased to inform you that your manuscript has been judged scientifically suitable for publication and will be formally accepted for publication once it meets all outstanding technical requirements.

Kind regards,

Andreas Zirlik, MD

Academic Editor

PLOS ONE
---

## [Editor Report · Acceptance letter]

4 Jul 2022

PONE-D-21-23252R2 

Regional and demographic variations of Carotid artery Intima and Media Thickness (CIMT): a systemic review and meta-analysis. 

Dear Dr. Abeysuriya:

I'm pleased to inform you that your manuscript has been deemed suitable for publication in PLOS ONE. Congratulations! Your manuscript is now with our production department. 

Kind regards, 

on behalf of

Univ. Prof. Dr. Andreas Zirlik 

Academic Editor

PLOS ONE